# Alginite-Rich Layers in the Bazhenov Deposits of Western Siberia

**Timur Bulatov** [1,*] , **Elena Kozlova** [1] , **Evgeniya Leushina** [1] , **Ivan Panchenko** [2] , **Natalia Pronina** [3] , **Andrey Voropaev** [4] , **Nikita Morozov** [5] **and Mikhail Spasennykh** [1]

1 Center for Hydrocarbon Recovery, Skolkovo Institute of Science and Technology (Skoltech), Skolkovo Innovation Center, Bolshoi Boulevard 30, Bld. 1, 121205 Moscow, Russia; e.kozlova@skoltech.ru (E.K.); e.leushina@skoltech.ru (E.L.); m.spasennykh@skoltech.ru (M.S.)
2 MiMGO, Entuziastov Highway 21, 111123 Moscow, Russia; panchenko@mimgo.ru
3 Faculty of Geology, Lomonosov Moscow State University, GSP-1, Leninskie Gory, 119991 Moscow, Russia; nvproncl@geol.msu.ru
4 HYDROISOTOP GmbH, Woelkestr. 9, 85301 Schweitenkirchen, Germany; av@hydroisotope.de
5 Gazprom Neft Science & Technology Centre, Moika River emb. 75-79 liter D, 190000 Saint Petersburg, Russia; morozov.nv@gazpromneft-ntc.ru
* Correspondence: timur.bulatov@skoltech.ru

**Abstract:** In this study, we identified the luminescent layers containing a significant amount of alginite in the Upper Jurassic–Lower Cretaceous Bazhenov Formation named "the alginite-rich layers". Lithological and geochemical methods were used to determine distinctive features of these layers and to evaluate their impact on the total petroleum generation potential of the Bazhenov Formation. We have shown that the composition of the alginite-rich layers differs significantly from the organic-rich siliceous Bazhenov rocks. Rock-Eval pyrolysis, bulk kinetics of thermal decomposition, elemental analysis, and the composition of pyrolysis products indicate type I kerogen to be the predominant component of the organic matter (OM). Isotope composition of carbon, nitrogen, and sulfur was used to provide insights into their origin and formation pathways. The luminescent alginite-rich layers proved to be good regional stratigraphic markers of the Bazhenov Formation due to widespread distribution over the central part of Western Siberia. They can also be applied for maturity evaluation of the deposits from immature to middle of the oil window, since the luminescence of the layers changes the color and intensity during maturation.

**Keywords:** the Bazhenov Formation; alginite-rich layers; macerals; type I kerogen; algal organic matter; marker horizon; Rock-Eval pyrolysis; kinetics; isotope composition





## 1. Introduction

Hydrocarbon production motivates the studies of unconventional reservoirs all over the world. The Bazhenov Formation discovered more than 50 years ago is considered to be one of the primary unconventional self-sourced reservoir in the world. Deposits of the Bazhenov Formation are extended over an area of about 1 million km$^2$ and are buried at depths from 2.0 to 3.5 km. In spite of a long research period and high number of studies, many questions still need to be addressed relating to effective technology of hydrocarbon exploration and production.

The Upper Jurassic–Lower Cretaceous Bazhenov Formation deposits have been formed in marine environments. A major part of solid organic matter in the Bazhenov Formation is represented by type II kerogen, which is characterized by a common marine genesis. The total organic carbon (TOC) concentrations vary considerably from less than 1 up to 30 wt.% for different lithological units within a cross-section. The TOC variations are usually accompanied by considerable variations of the hydrogen index (HI), chemical and isotope composition, and level of organic matter thermal maturation [1–4]. These

variations reflect the changes in sedimentation conditions, including the rate of sedimentation, diversity of marine biota and bioproductivity, redox conditions, and others [5–7]. In the case of the Bazhenov Formation, the effect of these factors on the hydrocarbon formation process and hydrocarbon productivity is not yet fully understood and requires additional studies, based on detailed analysis of OM composition, properties, and sources for different areas.

In this study, we focus on the layers containing luminescent organic matter characterized by high TOC values and extremely high HI values, distinguishing them from the other Bazhenov Formation rocks. We discovered these intervals in more than 20 wells located in the central parts of the Western Siberian petroleum basin [8] and performed their detailed lithological and geochemical analyses. The layers show bright yellow to orange luminescence under ultraviolet (UV) light, most probably due to their OM algal genesis. Their properties differ from other luminescent layers in the Bazhenov cross-sections, including oil saturated intervals and luminescent tuffs described in [9–11], which do not contain a high amount of organic carbon. Through a comprehensive analysis, we have proven the presence of type I kerogen in these layers, which is unique for the Bazhenov Formation. Therefore, an extensive study of these specific layers is important for understanding the sedimentation process in the Bazhenov paleobasin. The presence of luminescent layers in visually homogeneous Bazhenov cross-sections could become a tool for correlation of wells at least in the central part of Western Siberia where such layers have been found.

The identified alginite-rich layers were analyzed in terms of their lithological and geochemical characteristics, including their occurrence in the Bazhenov sequence, genesis, and potential application for well correlation and hydrocarbon exploration and production. This study considers the existing data on these types of layers found in the Bazhenov Formation and provides novel results based on a lithological investigation, Rock-Eval pyrolysis, CHNS analysis, stable isotope composition analysis, and other geochemical methods.

## 2. Geological Setting

The Bazhenov Formation and its facial and stratigraphical analogues Tutleimskaya (lower part), Maryanovskaya, Yanovstanskaya, and Golchikhinskaya Formations make up the single Bazhenov horizon that corresponds to the largest Mesozoic global black shale event. Figure 1 shows the Bazhenov Formation and indicates the locations of wells where the alginite-rich layers have been found.

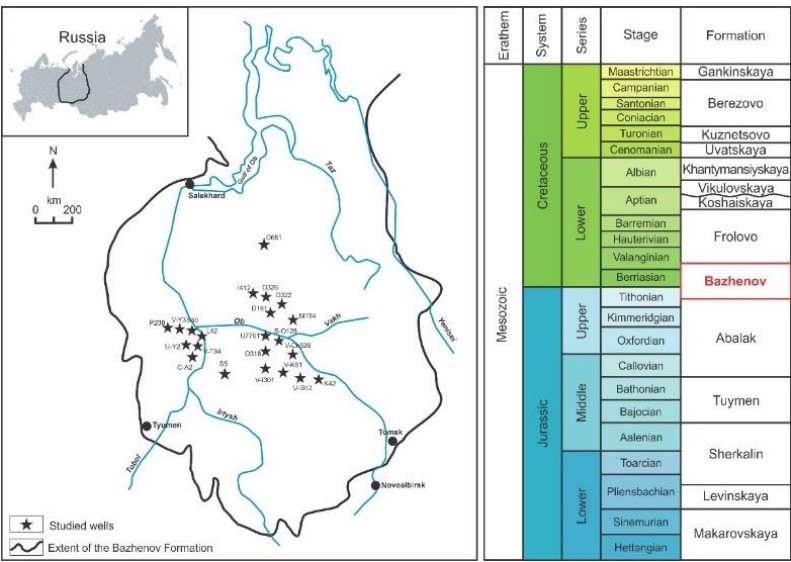

**Figure 1.** Map of the Bazhenov Formation with studied wells where the alginite-rich layers have been discovered (**left**) and stratigraphy of Mesozoic deposits of Western Siberia (**right**), modified from [12].

The sedimentation period covers the late Tithonian-early Valanginian time interval [13,14]. The Bazhenov epicontinental paleosea had extremely low deposition rates. High levels of bio-productivity and anoxic depositional environments were favorable for OM accumulation and preservation [15–17]. Numerous studies have integrated the paleontological findings and other proxies of the paleoenvironment conditions that controlled its deposition at different stages of geological history [18–21]. In general, the Bazhenov Formation rock properties, organic matter, and their transformation into petroleum have been the focus of numerous studies since the 1960s [22–29].

The stratigraphic units of the Bazhenov Formation are not unified, and different studies have subdivided them according to different criteria. Nonetheless, most studies accept the division of the Bazhenov Formation into an upper part and a lower part due to lithological characteristics and concentration of OM, which is higher in the Upper Bazhenov. In this study, we adopt the stratigraphic units reported in [30]. According to Panchenko et al., the Bazhenov Formation is divided into two parts, each consisting of three distinct units. Thus, the Bazhenov section is comprised of six units, possessing certain paleontological, lithological, and geochemical characteristics, which have been identified based on well logging data (Figure 2). Such an integrated approach allows one to establish units that reflect event-driven changes in the sedimentation conditions during the late Tithonian-early Valanginian period.

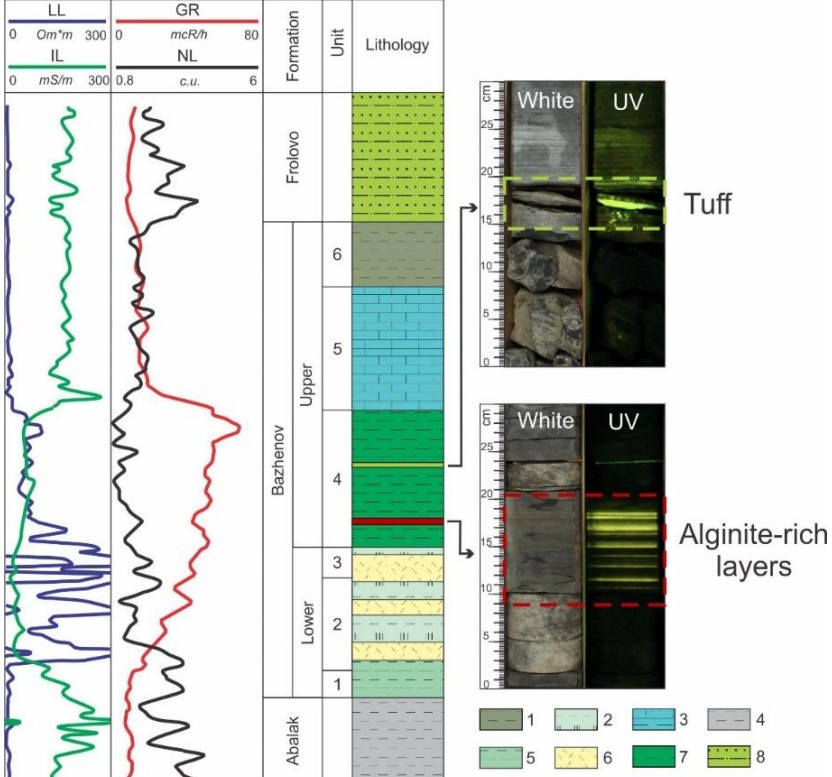

**Figure 2.** Example of the Bazhenov Formation cross-section. Core photographs under white and UV light. Lithology: 1, siliceous claystone; 2, siliceous rocks with low clay content; 3, organic-rich calcareous rocks; 4, claystone; 5, organic-rich claystone; 6, radiolarite; 7, organic-rich siliceous rocks; 8, alternation of siltstones, claystone, and sandstones. Well-logs: LL, laterolog; IL, induction log; GR, gamma-ray log; NL, neutron log.

The Bazhenov section is mainly composed of organic-rich siliceous rocks with varying admixtures of clay and carbonate minerals. Different quantities of siliceous rocks, which are composed of radiolarites and carbonate rocks (limestones and dolomites), can be found in every cross-section of the Bazhenov Formation. The rock-forming minerals are quartz,

clay (mixed-layer, kaolinite, and chlorite), and carbonate (calcite and dolomite) minerals. Sulfides are disseminated within the deposits predominantly in a form of pyrite. Pyrite can reach high concentrations up to 15 wt.%. Apatite and glauconite are also observed within thin lenses and layers.

Similar to other black shales deposited in marine environments, the Bazhenov Formation rocks contain a higher concentration of transitional metals as compared with ordinary shales, for example, Ni, Co, Cd, Mo, Ba, V, Cu, Zn, U, and others. The Bazhenov Formation is of particular interest due to its high uranium content [15,17,31], which is detected as an anomaly in gamma-ray logs [32].

The Bazhenov Formation organic matter, its generalized structure, and composition have been analyzed in a range of previous studies [33–37]. Total organic carbon concentration varies from 1 to 25–30 wt.% (mean 8–9 wt.%) and the initial hydrogen index ($HI_0$) values range between 650 and 715 mg HC/g TOC [38–40]. The organic matter is type II kerogen with maturity varying along the region from immature (protocatagenesis stage PC3) in the Nyurolskaya Depression to the end of the oil window (mesocatagenesis stage MC3) in the Salym Arch [39,41,42]. Catagenesis stages are named according to [43], tectonic zoning is suggested by [44]. Type I kerogen in the Bazhenov deposits has been mentioned previously in recent studies, but the detailed characterization has never been presented [38,39].

Depending on the thermal maturity advance, the properties and characteristics of the OM in the Bazhenov Formation change, which are reflected in the Rock-Eval pyrolysis parameters [45–47], molecular, and isotope composition [4,48,49]. In general, areas with a higher maturity of OM show better hydrocarbon productivity.

Luminescent alginite-rich layers are located in the lower part of the Upper Bazhenov, in Unit 4. There are other luminescent layers present in Unit 4, i.e., tuffs (Figures 2 and 3). Tuffs appear as highly fractured greenish-gray or brown rocks with a relict volcaniclastic microstructure. They are mostly composed of devitrified volcanic glass, which has been subjected to hydration activity. The alginite-rich layers and tuffs have a different nature of luminescence. Various studies have associated the luminescence of tuffs with the mineralogy of clay minerals [10] or barite concentrations [11], whereas the alginite-rich layers have an organic nature of luminescence. Since the luminescence of the alginite-rich layers and tuffs are very similar visually, in the results section we provide their comparative characteristics in order to determine the strict criteria to distinguish them from one another.

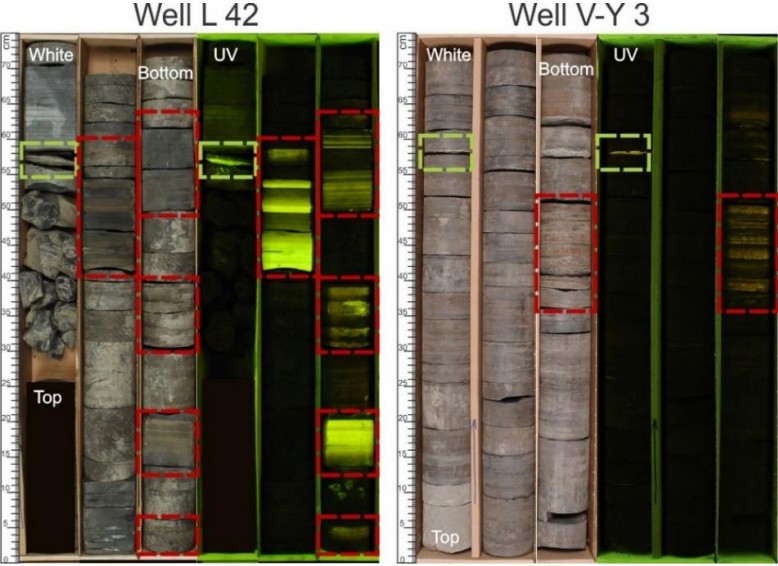

**Figure 3.** Core layout showing typical appearance of the Bazhenov Formation organic-rich siliceous rocks with presence of the alginite-rich layers (marked in red) and tuffs (marked in light green). Core photographs under white and UV light.



## 3. Materials and Methods

Luminescent alginite-rich layers were observed in the core material from 22 wells (Figure 1). All of the alginite-rich layers occur in the lower part of the Upper Bazhenov Formation (Unit 4, see Figure 2).

For this study, we have chosen samples of the alginite-rich layers and host rocks representing mostly organic-rich siliceous rocks. The sampling distance between the alginite-rich layers and the host rocks did not exceed 5 cm.

The samples for thin section petrographic examination were cut perpendicular to the bedding planes, oriented down-up, 0.03 mm thick, polished, and cover slipped. The thin sections were examined using Axioscop 5 polarizing microscope (Zeiss, Jena, Germany) equipped with an Axiocam 506 color digital camera (Zeiss, Jena, Germany).

Polished whole-rock blocks were prepared and examined. The maceral analysis was performed using a QDI-300 Craic micro-spectrophotometer (CRAIC Technologies, San Dimas, USA) with a DM 2500 P base microscope (Leica, Wetzlar, Germany) under reflected white light and UV light illumination in air [50]. A quartz halogen lamp (12 V, 100 W) was used for white light. A high-pressure mercury lamp (100 W HBO®, OSRAM, Munich, Germany) was used for the UV light. The maceral nomenclature used in this study follows the International Committee for Coal and Organic Petrology [51–53].

The X-ray diffraction (XRD) analysis was performed on selected samples for bulk mineral composition analyses. The XRD analyses were carried out with ARL X'TRA X-ray diffractometer (Thermo, Basel, Switzerland) using CuK$\alpha$ radiation, operated at 40 kV and 40 mA and analyzed over the 2–60° 2$\theta$ angular range. For indexing of diffraction peaks and identification of mineral phases, the Siroquant software package (Siroquant, Mitchell, Australia) was used. We used the Rietveld refinement procedure and obtained data through a concerted theory-experiment comparison.

The X-ray fluorescence (XRF) analysis was performed using ARL Perform'X spectrometer (Thermo, Basel, Switzerland), applying the fundamental parameters method. The samples were prepared as pressed powder pellets on a boric acid substrate.

Rock-Eval pyrolysis was performed using a HAWK Resource Workstation (Wildcat Technology, Humble, USA). We determined the following main pyrolysis parameters: thermovaporized free hydrocarbons (S0 and S1), pyrolysis products from cracking of kerogen and heavy petroleum fractions (S2), $CO_2$ generated from OM during the pyrolysis step (S3), $CO_2$ generated from organic matter during the oxidation step (S4), $CO_2$ generated from the mineral source during the oxidation step (S5), and the temperature at which the maximum amount of hydrocarbon was generated ($T_{max}$). Above parameters were used to calculate the TOC, generative organic carbon (GOC), non-generative organic carbon (NGOC), hydrogen index (HI), oxygen index (OI), production index (PI), and proportion of generative kerogen ($K_{goc}$ = GOC/TOC × 100).

The kinetic studies of organic matter thermal decomposition were carried out using a HAWK Resource Workstation. The extracted rock samples were subjected to non-isothermal pyrolysis at temperature from 300 to 650 °C, at three heating rates (3, 10, and 30 °C/min). The Kinetics 2015 software (CoeoIsoChem Corporation, Covina, USA) was used to determine the discrete distribution of activation energies ($E_a$, kcal/mol) for a fixed frequency factor $A = 1 \times 10^{14}$ s$^{-1}$.

The CHN628 elemental analyzer (LECO, St. Joseph, MO, USA) was used to determine the carbon, hydrogen, and nitrogen content via combustion (at 950 °C) into $CO_2$, $H_2O$, and $NO_2$ contents, respectively. A helium carrier gas swept the combustion gas to separate infrared cells utilized for the detection of $H_2O$ and $CO_2$, while a thermal conductivity cell was used for the detection of nitrogen. The 628 S module (LECO, St. Joseph, MO, USA) was used to determine the sulfur content. Sulfur evolved (at 1350 °C) from the sample and formed $SO_2$. From the combustion system, the gases flowed through the sulfur infrared detection cell. Combustion occurred in the pure oxygen for CHN and S analyses.

Pyrolysis gas chromatography analyses (pyro-GC × GC-FID/TOFMS) were performed using the Pegasus 4D (LECO, St. Joseph, MO, USA) equipped with a cryo-

modulator and injection modules for thermal desorption and pyrolysis (Gerstel, Mulheim, Germany). Chromatographic separation of the products was carried out using a reverse-order column set. Samples were heated to a final temperature of 500 °C. Pyrolysis products were detected simultaneously using FID and time-of-flight mass spectrometer detectors.

Isotope and elemental analysis of carbon, nitrogen, and sulfur was performed on a DELTA V Plus mass spectrometer (Thermo, Dreieich, Germany) equipped with a Flash HT elemental analyzer (Thermo, Dreieich, Germany). The validation of analytical procedures was established using the following international standards: oil NBS 22, ammonium sulfate IAEA-N-2, and barium sulfate NBS 127 for carbon, nitrogen and sulfur, respectively, as well as the laboratory standards, whose elemental and isotope composition was determined during interlaboratory comparison measurements. In accordance with the generally accepted rules, all the measured isotopic ratios were recalculated and are given in δ values characterizing the deviation of the isotopic ratio of the sample from the isotopic ratio of the standard (in ‰), i.e., PDB, AIR and CDT for carbon, nitrogen and sulfur, respectively. The absolute error of measurement for δ values is ±0.3‰ for carbon, ±0.5‰ for sulfur and nitrogen. The relative error of element content measurements is ±10%.

## 4. Results

### 4.1. Visual Core Description under White and UV Light

The thicknesses of the individual alginite-rich layers vary from 1 to 50 mm, and the total thickness is up to 1 m in the cross-section. The studied cross-sections of the Bazhenov Formation have a thickness of about 20–30 m. The alginite layers usually constitute about 1–5% of the entire cross-section. They can be macroscopically distinguished by their bright yellow or orange luminescence under UV light and less often by lighter coloring as compared with the darker host rocks (organic-rich siliceous rocks) under white light (Figure 3). Usually, the alginite-rich layers pass gradually to the organic-rich siliceous rocks. This gradation gives the rocks a banded appearance under UV light. Another interesting feature of the alginite-rich layers is their plasticity and flexibility, as well as their low density.

### 4.2. Lithology and Mineral Composition

According to petrographic description, the luminescent alginite-rich layers are mainly composed of OM and quartz aggregations (Figure 4a,b). Remains of fish bones, calcispheres, and radiolarians are rare. Pyrite is present either in the form of framboids, evenly distributed over the rock, or replaces radiolarian shells. The boundaries between the alginite-rich layers and the underlying rocks are sharp (Figure 4c), while the upper contacts are gradually replaced by the overlying rocks.

From the XRD results, it is clear that the mineral part of the alginite-rich layers mainly consists of quartz (70–90 wt.%). The content of clay minerals, including kaolinite and mixed-layer clays, usually does not exceed 17 wt.%. Carbonate minerals and pyrite are present in minor amounts. The results of the XRF confirm the mineral composition data and indicate a predominance of the siliceous component ($SiO_2$ values range 65–90 wt.%) in an inorganic part of the alginite-rich layers.

The host rocks are organic-rich siliceous rocks. Thin sections show lamination of organic-rich rock, generally pyritic, with organic matter along the bedding axis (Figure 4d). This rock is generally non-calcareous. Pyrite occurs as microcrystals, framboidal aggregates, or as nodules formed within the rocks. The organic-rich siliceous rocks contain rare phosphate fish bone remains oriented parallel to the layering.

The XRD shows that a major part of organic-rich siliceous rocks is quartz (60–70 wt.%) in addition to clay minerals (15–20 wt.%), calcite (0–10 wt.%), and pyrite (3–10 wt.%). The most abundant oxides are $SiO_2$, $Al_2O_3$, CaO, and $Fe_2O_3$, whereas MgO, $TiO_2$, MnO, $K_2O$, $P_2O_5$, and $Na_2O$ are present in minor quantities. For hosting organic-rich siliceous rocks, the $SiO_2$ values are in the range of 50–70 wt.%, the $Al_2O_3$ values are in the range of

5–15 wt.%, and the CaO values are in the range from 0.5 to 15 wt.%. The average $Fe_2O_3$ content is 4.6 wt.%.

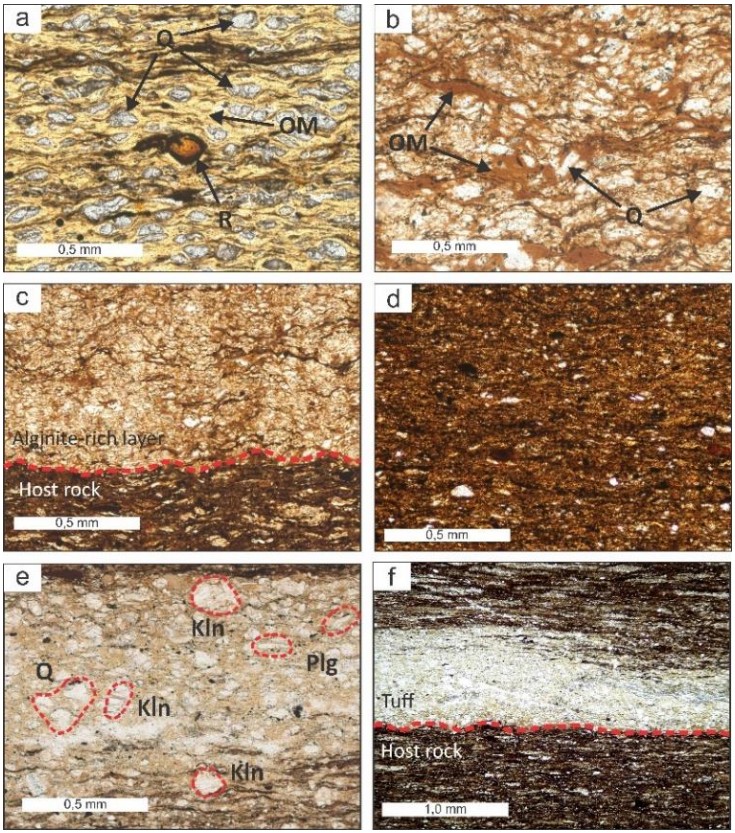

**Figure 4.** Typical thin section photomicrographs of the alginite-rich layers, host rocks, and tuffs. (**a**,**b**) Alginite-rich layers, predominantly containing organic matter (OM) and aggregates of quartz (Q), radiolarians (R) are rare; (**c**) alginite-rich layer and host rock that are separated by wavy surfaces (highlighted by red short-dashed line); (**d**) host rock shows the distribution of mineral matrix with organic matter along beddings; (**e**) tuff shows former glassy ash matrix devitrified to crystals of plagioclase (Plg), quartz (Q), kaolinite (Kln), and clay matrix; (**f**) host rock contains thin tuff layer. All photomicrographs are taken under parallel nicols. Corresponding magnification of (**a**–**e**) 20× and (**f**) 10×.

For comparison, we present photomicrographs of the thin sections of the luminescent tuffs (Figure 4e,f). The tuffs consist of volcanic glasses, almost completely transformed into clay minerals. In the petrographic thin sections, we observed plagioclase and zircon grains. The tuffs are significantly different from the alginite-rich layers and from the host siliceous rocks by the almost complete absence of OM and by a substantially clay mineral composition.

According to the XRD, tuffs are essentially composed of clay minerals (up to 60–80 wt.%). The predominant clay minerals of all the tuffs are mixed-layer clays and kaolinite. The minor minerals are quartz, plagioclase, mica, albite, and pyrite. The XRF results of tuffs show that $SiO_2$ values range between 45 and 55 wt.%. Similarly, $Al_2O_3$ contents vary between 20 and 30 wt.%. The average $Fe_2O_3$ content is 2.4 wt.% and that of CaO is 1.2 wt.%. The sum of $Na_2O$ and $K_2O$ is 3–6 wt.%. The chemical composition of tuffs corresponds to the basaltic andesitic source of intermediate magma.

*4.3. Organic Petrography*

The organic petrography shows that alginite is a major organic component. The shape of the OM has fan, lumps, and clots morphology. Alginite has a distinctive external form

and, in most cases, an internal structure of specific recognizable algal remains. Under reflected white light, alginite is brownish (Figure 5a,c,e). During the analysis in the incident fluorescent mode (under UV light), alginite is a greenish-yellow color to bright yellow color (Figure 5b,d,f). Therefore, alginite is separated from the amorphous material, i.e., bituminite, which lacks distinctive morphology and originates from the various precursors.

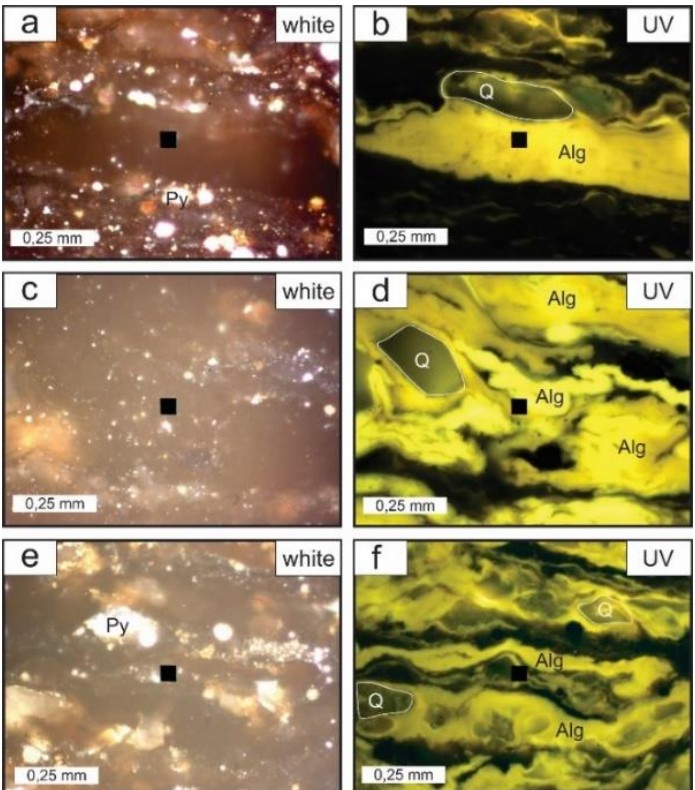

**Figure 5.** Photomicrographs illustrating a typical maceral composition of the alginite-rich layers. (**a,c,e**) Samples under reflected white light shows enrichment of organic matter in mineral matrix. Many colonies contain pyrite; (**b,d,f**) same samples under UV light shows high intensity yellow luminescence of alginite. Air, magnification of 50× for all photomicrographs. Alg, alginite; Q, quartz; Py, pyrite.

The absolute predominance of alginite (40–50 vol.%) in the maceral composition confirms the algal origin of the organic matter in these layers. As compared with the samples from the host rocks, alginites are less abundant and only small amounts of bituminite were detected.

### 4.4. Characterization of Organic Matter

Since the petrographic data allowed us to state that OM in the alginite-rich layers had an algal nature, we further analyzed the quantity and composition of the OM to prove this hypothesis. The geochemical characterization included Rock-Eval pyrolysis, pyro-GC-TOF MS/FID analyses, kinetic studies of OM thermal decomposition, elemental analysis, and isotope composition measurements.

#### 4.4.1. Rock-Eval Pyrolysis

In Table 1, we report the Rock-Eval pyrolysis data for the studied samples from the alginite-rich layers and host rocks. To emphasize the difference, we provide Rock-Eval pyrolysis data for tuffs sampled during the study.

**Table 1.** Rock-Eval pyrolysis data for the alginite-rich layers, host rocks, and tuffs.

| Well | Sample | S0 + S1, mg HC/g | S2 mg HC/g | PI (S1/S1 + S2) | TOC, wt.% | $K_{goc}$, % | $T_{max}$, °C | HI, mg HC/g TOC | OI, mg $CO_2$/g TOC |
|------|--------|------------------|------------|------------------|-----------|--------------|----------------|------------------|----------------------|
| V-K 81 | Alginite-rich layer | 2.15 | 241.65 | 0.01 | 22.94 | 91 | 445 | 1053 | 1 |
| | Host rock | 4.64 | 133.90 | 0.03 | 17.81 | 67 | 431 | 751 | 4 |
| V-Ch 526 | Alginite-rich layer | 1.06 | 219.02 | 0.01 | 21.56 | 87 | 443 | 1015 | 1 |
| | Host rock | 5.12 | 74.52 | 0.06 | 10.72 | 67 | 430 | 682 | 3 |
| M 184 | Alginite-rich layer | 0.53 | 259.24 | 0.00 | 25.25 | 88 | 451 | 1026 | 3 |
| | Host rock | 1.36 | 46.21 | 0.03 | 8.77 | 48 | 438 | 527 | 11 |
| S-O 128 | Alginite-rich layer | 2.49 | 110.48 | 0.02 | 12.20 | 79 | 450 | 905 | 6 |
| | Host rock | 2.99 | 71.05 | 0.04 | 13.20 | 49 | 444 | 538 | 4 |
| O 318 | Alginite-rich layer | 1.36 | 103.14 | 0.53 | 12.19 | 74 | 447 | 846 | 4 |
| | Tuff | 0.22 | 2.08 | 0.10 | 0.77 | 34 | 436 | 269 | 36 |
| | Host rock | 5.65 | 66.43 | 0.45 | 13.9 | 48 | 436 | 478 | 3 |
| S 40 | Alginite-rich layer | 1.79 | 140.25 | 0.01 | 16.21 | 76 | 440 | 865 | 2 |
| | Host rock | 2.81 | 51.93 | 0.05 | 10.69 | 46 | 430 | 485 | 3 |
| V-I 301 | Alginite-rich layer | 0.50 | 66.78 | 0.01 | 8.19 | 72 | 439 | 815 | 4 |
| | Host rock | 3.44 | 54.25 | 0.06 | 11.03 | 48 | 431 | 491 | 5 |
| D 541 | Tuff | 0.16 | 1.88 | 0.08 | 0.71 | 28 | 454 | 264 | 28 |
| M 14 | Tuff | 0.25 | 2.03 | 0.11 | 0.60 | 37 | 421 | 339 | 59 |
| J 177 | Tuff | 0.34 | 1.89 | 0.15 | 0.79 | 28 | 437 | 240 | 29 |

The most distinctive feature of organic matter of the alginite-rich layers is the value of hydrogen index. HI in the alginite-rich layers of a low maturity degree reaches 1053 mg HC/g TOC, which is 350–400 mg HC/g TOC higher than that for host rocks. In the modified van Krevelen diagram, the alginite-rich layers are located in the area that is typical for type I kerogens, while the Bazhenov Formation OM of the host rocks is characterized as type II kerogen (Figure 6a). The TOC concentration of the low mature alginite-rich layers reaches 25 wt.%. The amount of pyrolyzed hydrocarbon compounds in the studied intervals reaches 260 mg HC/g rock. In the S2-TOC plot (Figure 6b), the alginite-rich layers have significantly better petroleum generation potential than the host rocks [54,55]. There is a minor difference in oxygen index between the alginite-rich layers and host rocks (1–11 mg $CO_2$/g TOC). At the same time, in tuffs, OI sharply increases and reaches 59 mg $CO_2$/g TOC. The maturity parameters of the alginite-rich layers are inconsistent with each other. At very low PI we observe very high $T_{max}$ values. These values exceed the corresponding values in the host rocks by 6–14 °C, which reflect the differences in the OM maturation pathways for different kerogen types, provided the OM type remains comparable [56].

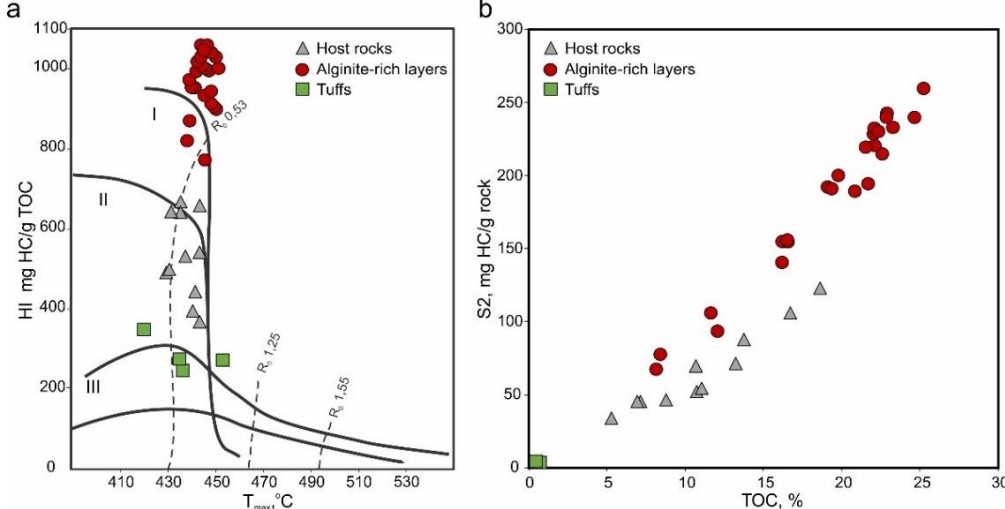

**Figure 6.** Modified van Krevelen diagram (**a**) and S2 vs. TOC plot (**b**) for the studied samples.

Organic matter in the alginite-rich layers has a higher quantity of pyrolyzable carbon as compared with the background along the section. Generative organic carbon to TOC ratio ($K_{goc}$) reaches 91% and decreases with an increase in OM maturity (Table 1).

Figure 7 illustrates a section of a core sample (7 cm thick) with alternating layers of bright and dull luminescent intensities, related to alginite contents. The samples were analyzed by Rock-Eval pyrolysis, and we observed significant variations of the parameters even in such a small interval.

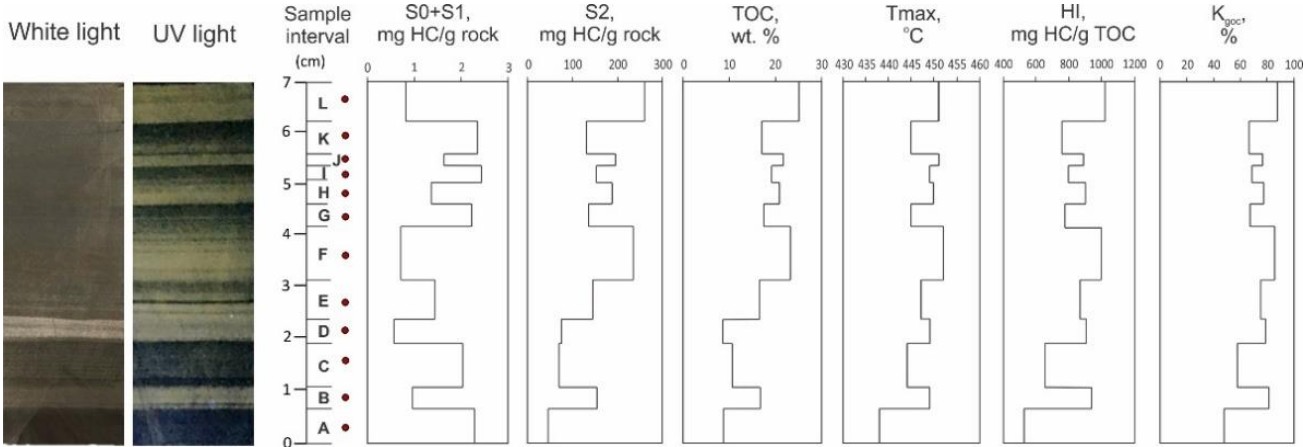

**Figure 7.** Detailed (layer-by-layer) Rock-Eval pyrolysis for core sample with interbedding of the alginite-rich layer and hosting organic-rich siliceous rock from well V-K 81. A–L, sampling points.

Rock-Eval pyrolysis parameters obtained for the first background Point A (Figure 7) are common for the Bazhenov Formation rocks; for darker non-luminescent layers, the TOC is 8–11 wt.%; and for the luminescent alginite-rich layers, the TOC is 13–25 wt.%. For the luminescent layers, we observe a significant increase in $T_{max}$ by 5–11 °C more than the darker ones. The difference in HI for the nearby layers reaches 450 mg HC/g TOC. The amount of thermally desorbed light hydrocarbons (S0 + S1) changes inversely to S2 and TOC, i.e., for luminescent rocks S2 is high and S0 + S1 is low, and vice versa for the darker layers. Thus, the layer-by-layer Rock-Eval pyrolysis emphasizes the heterogeneous saturation of the Bazhenov Formation rocks with type I kerogen, which interbeds and mixes with different types of OM (type I and type II of kerogen).

4.4.2. Organic Matter Thermal Decomposition

The presence of type I kerogen in the alginite-rich layers is confirmed by the bulk kinetic characteristics of the OM thermal decomposition. In Figure 8, we provide the activation energy distributions for two pairs of samples, i.e., from the alginite-rich layer and from the adjacent host rock. The spectrum for the alginite-rich organic matter is narrow and consists of a single energy Ea = 53 kcal/mol (Figure 8b,d), which is a feature of type I kerogen [57,58]. For the host rocks, the activation energy distributions (with a maximum at 52 kcal/mol) have the typical shape of the Bazhenov kerogen and characterize the host rocks OM as type II kerogen (Figure 8a,c). The upper pair of samples (Figure 8a,b) was taken from the section of the Bazhenov Formation with moderately immature OM (MC1). The lower one (Figure 8c,d) was taken from the section in the early oil window (MC1-2).

The results of the pyrolysis-gas chromatography/mass spectrometry provide us with information about the quantitative and qualitative chemistry of the thermal decomposition products from the cracking of kerogen [59]. This method provides a direct indicator for the geochemical typing of kerogens and types of hydrocarbons that can be generated during thermal maturation. A comparison of the aliphatic and aromatic nature of kerogens has been previously described in [57,59–61].

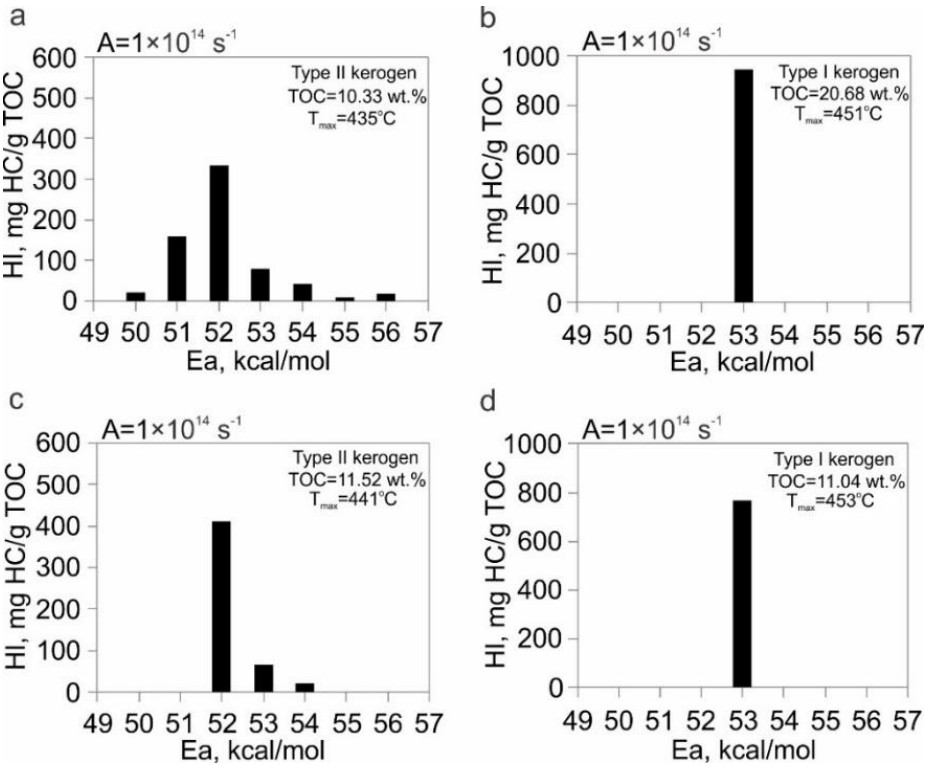

**Figure 8.** Activation energy distribution of organic matter thermal decomposition for the host rocks (**a**,**c**) and for the alginite-rich layers (**b**,**d**). Frequency factor A is fixed to $1 \times 10^{14}$ s$^{-1}$.

Numerous compound hydrocarbon classes have been found in pyrolysis products of kerogen, including light hydrocarbons C1–C7; n-alkanes C8–C40; unsaturated HC; naphthenes; and mono-, di-, tri- and poly-aromatic compounds (Table 2). Type I kerogen of the alginite-rich layers produces dominantly long-chain n-alkanes and n-alkenes, whereas type II kerogen of host rocks produces more unsaturated HC, naphthenes, and aromatic compounds (Figure 9a,b). These differences are also reflected in the elemental composition. The type I kerogen of the alginite-rich layers has a high atomic H/C up to 1.88 and a low atomic O/C (0.05–0.07); the kerogen of host rocks characterized by a high atomic H/C (1.09–1.19) and a low atomic O/C (0.06–0.09).

**Table 2.** Distribution of HC classes from Pyro-GC×GC-TOF MS/FID and elemental analysis data for the alginite-rich layers and host rocks.

| Hydrocarbon Classes (% *w/w*) | Well S-O 128 | | Well V-I 301 | |
| --- | --- | --- | --- | --- |
| | **Host Rock** | **Alginite-Rich Layer** | **Host Rock** | **Alginite-Rich Layer** |
| Light HC C1–C7 | 7 | 6 | 19 | 8 |
| n-Alkanes C8–C40 | 8 | 25 | 25 | 33 |
| Unsaturated HC and naphthenes | 37 | 64 | 29 | 53 |
| Mono-aromatic compounds | 21 | 4 | 19 | 4 |
| Di-aromatic compounds | 15 | 1 | 7 | 1 |
| Tri- and poly-aromatic compounds | 12 | 0 | 1 | 1 |
| Aliphatic/aromatic | 52/48 | 95/5 | 73/27 | 94/6 |
| H/C | 1.09 | 1.88 | 1.19 | 1.69 |
| O/C | 0.06 | 0.07 | 0.09 | 0.05 |

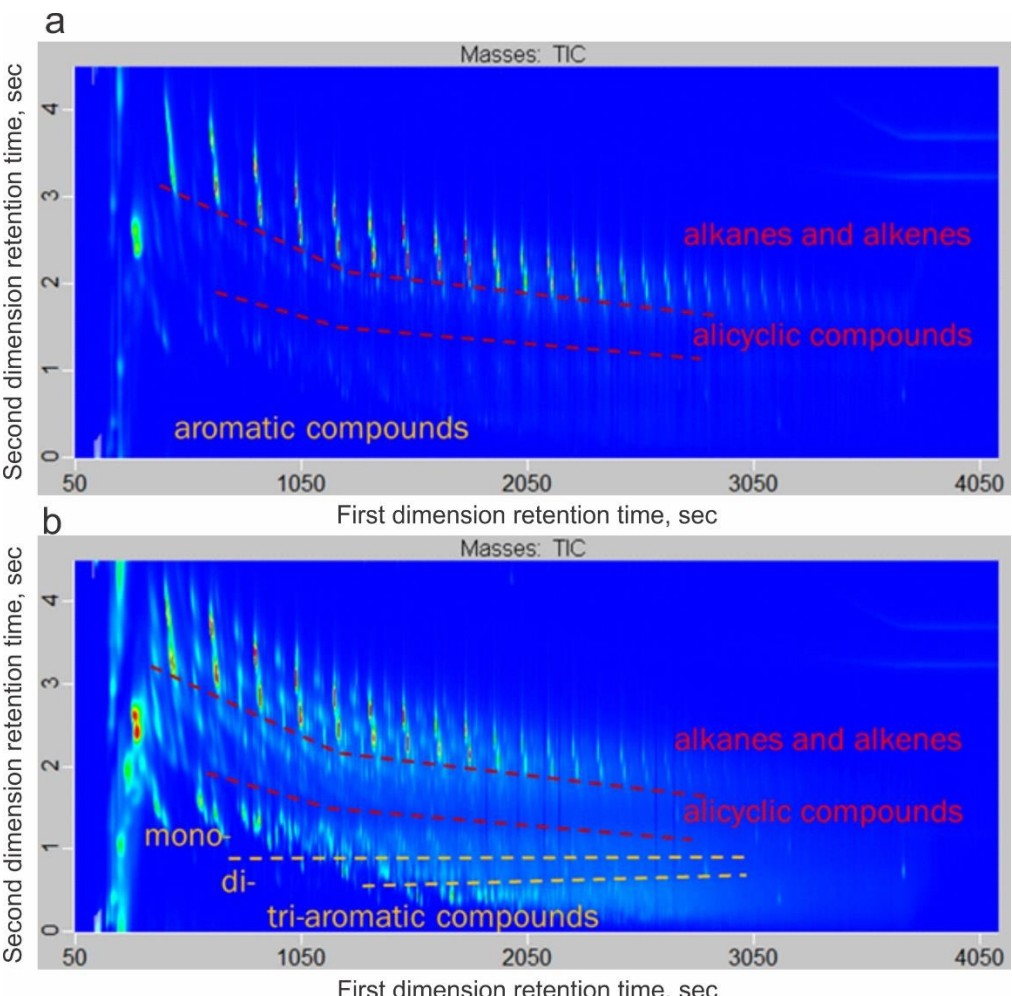

**Figure 9.** Two-dimensional total ion chromatograms (TIC) of kerogen cracking products for the alginite-rich layers (**a**) and host rocks (**b**).

High aliphaticity (at least 90 wt.%) of pyrolysis products suggest that type I kerogen of the alginite-rich layers is dominated by lipid-rich and protein-rich algal material that has undergone extensive bacterial reworking in reducing conditions. Aliphatic to aromatic ratio of pyrolysis products show that type II kerogen of host rocks originates from various precursors (mixed phytoplankton, zooplankton, and bacterial material).

### 4.4.3. Isotope Composition of C, N, and S

Among geochemical data, the isotope composition of carbon, nitrogen, and sulfur in rocks enables the genesis estimation of organic matter. In Table 3, we provide the results of stable isotope studies of carbon, nitrogen, and sulfur.

The carbon isotope composition for the alginite-rich layers varies in the range from −31.1 to −31.9‰, while for the host rocks $\delta^{13}C_{org}$ it varies from −30.5 to −31.9‰. In contrast, tuffs contain 10–30 times less organic carbon than host rocks and the alginite-rich layers. Tuffs are enriched in heavy carbon isotope, and for $\delta^{13}C_{org}$ it is from −29.0 to −28.8‰.

The concentrations of nitrogen in the alginite-rich layers and in host rocks vary from 0.07 to 0.37 wt.%. For all the samples from the alginite-rich layers, we observe 1.5–2 times lower nitrogen content as compared with the host rocks. The concentrations of nitrogen in the host rocks vary from 0.24 to 0.37 wt.%, and in $\delta^{15}N_{tot}$, vary from 0.9 to 5.4‰. The nitrogen contents in the alginite-rich layers is from 0.07 to 0.21 wt.%, and the organic matter is enriched in $^{15}N$ (in $\delta^{15}N_{tot}$, it is from 5.9 to 25.7‰). The $C_{org}/N_{tot}$ ratio correlates

with the heavier nitrogen isotope composition (Figure 10a). The tuffs are characterized by $C_{org}/N_{tot}$ ratios from 2.7 to 4.38, and by $\delta^{15}N_{tot}$ values from 1.2 to 2.4‰.

**Table 3.** Bulk C, N, and S elemental and isotope composition for the alginite-rich layers, host rocks, and tuffs.

| Well | Sample | $C_{org}/N_{tot}$ | $\delta^{13}C_{org}$, ‰ PDB | $N_{tot}$, wt.% | $\delta^{15}N_{tot}$, ‰ AIR | $S_{tot}$, wt.% | $\delta^{34}S_{tot}$, ‰ CDT | $C_{org}$, wt.% |
|---|---|---|---|---|---|---|---|---|
| V-K81 | Alginite-rich layer | 93.37 | −31.5 | 0.21 | 5.9 | 2.31 | −6.0 | 22.12 |
| | Host rock | 43.84 | −31.4 | 0.37 | 0.9 | 4.36 | −17.3 | 17.81 |
| S-O 128 | Alginite-rich layer | 103.28 | −31.8 | 0.11 | 7.7 | 1.61 | −18.5 | 12.20 |
| | Host rock | 43.54 | −30.5 | 0.34 | 3.2 | 3.71 | −24.9 | 12.30 |
| V-I 301 | Alginite-rich layer | 109.57 | −31.8 | 0.07 | 17.5 | 2.75 | −11.2 | 8.19 |
| | Host rock | 42.23 | −31.9 | 0.24 | 5.4 | 4.18 | −26.0 | 11.03 |
| V-C 526 | Alginite-rich layer | 155.50 | −31.1 | 0.15 | 25.7 | 1.37 | −9.3 | 21.56 |
| | Host rock | - | - | - | - | - | - | - |
| L 42 | Alginite-rich layer | 76.83 | −31.9 | 0.21 | 13.0 | 2.10 | −7.1 | 19.79 |
| | Host rock | - | - | - | - | - | - | - |
| D 541 | Tuff | 2.70 | −28.8 | 0.45 | 2.4 | 1.89 | 4.4 | 0.71 |
| M 14 | Tuff | 4.38 | −29.0 | 0.21 | 1.5 | 0.69 | 4.5 | 0.60 |
| J 177 | Tuff | 4.11 | −28.9 | 0.23 | 2.1 | 0.63 | 2.4 | 0.79 |

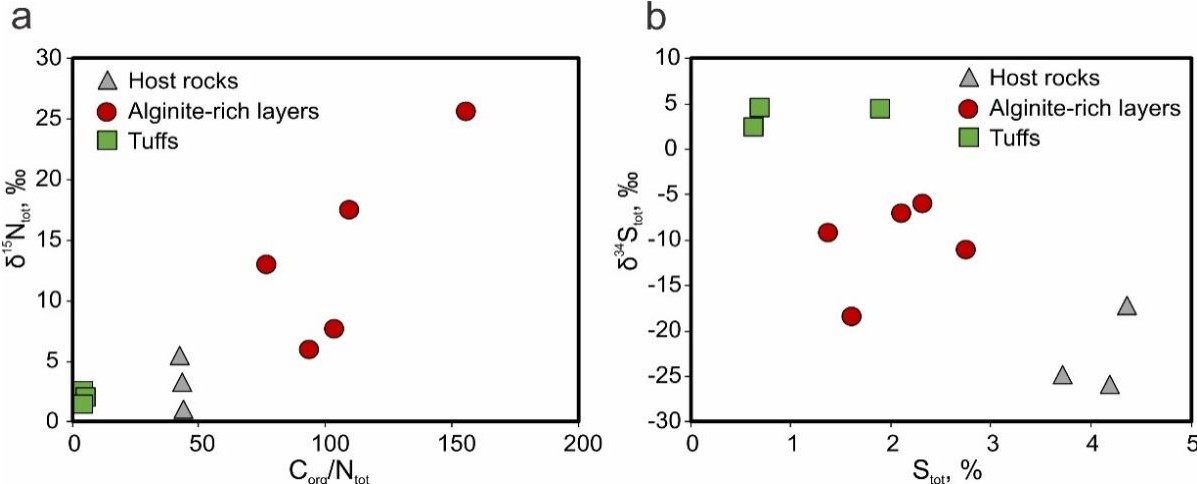

**Figure 10.** Comparison of $\delta^{15}N_{tot}$ values and $C_{org}/N_{tot}$ ratio (**a**), and comparison of $\delta^{34}S_{tot}$ values and $S_{tot}$ content (**b**) in the alginite-rich layers, host rocks, and tuffs.

Content and isotope composition of the total sulfur is also different for the rocks from the alginite-rich, host deposits, and tuff. For the alginite-rich layers, the $\delta^{34}S_{tot}$ values range from −6.0 to −18.5‰ with a total sulfur content of 1.37 to 2.75 wt.%. For the host rocks, we detect a predominance of lighter sulfur, i.e., the $\delta^{34}S_{tot}$ values varies from −17.3 to −26.0‰ and for $S_{tot}$, from 3.71 to 4.36%. Tuffs demonstrate the heaviest sulfur isotope ratio (Figure 10b).

## 5. Discussion

### 5.1. Occurrence

Alginite-rich layers are discovered throughout the Bazhenov sequence within the central part of Western Siberia in Unit 4, where the Jurassic–Cretaceous boundary is situated between the Upper and Lower Bazhenov Formation. The remains of variable-sized algae are regularly microscopically recorded in rock samples, particularly in fine-grained organic-rich siliceous rocks that occur throughout the Bazhenov sequence, but pure and

concentrated fossil algae are found only in the alginite-rich layers. According to the results, the alginite-rich layers widely extend for more than 455,000 km$^2$ area (Figure 1). The studied wells are located in different regions of Western Siberia and are confined to the following various tectonic elements: the Frolovskaya Megadepression, the Krasnoleninsky Arch, the Surgut Arch, the Nizhnevartovsky Arch, the Yugansk Megadepression, and the Nyurolskaya Depression and others.

### 5.2. Key Characteristics of the Alginite-Rich Layers

Considering visual characteristics and the obtained mineralogical and geochemical data, the key features that can be used for identifying the alginite-rich layers from host rocks are: macroscopically visually lighter coloring as compared with the darker host rocks; luminescence under UV light; low density and plasticity; matrix consists mainly of algal OM and quartz aggregations; Rock-Eval pyrolysis data, kinetics parameters, and elemental composition indicate type I kerogen in the alginite-rich layers; high $C_{org}/N_{tot}$ ratio and $\delta^{15}N_{tot}$ values.

The Rock-Eval pyrolysis parameters of the alginite layers significantly exceed those in host rocks. The values of $K_{goc}$ reach 90%. The hydrogen indices are 400–500 mg HC/g TOC higher, and the $T_{max}$ is higher by 5–10 °C as compare with the host rocks. The atomic H/C ratio is up to 1.88. The Rock-Eval pyrolysis parameters, bulk kinetics data, and elemental composition confidently describe OM as type I kerogen.

The mineral composition of alginite-rich layers mainly consists of quartz grains (60–90 wt.%), which is not typical for the Bazhenov Formation rocks. In general, the abnormal quartz concentrations within Unit 4 can indirectly show the occurrence of the alginite-rich layers.

The carbon isotope composition of OM indicates the source of organic matter. The $\delta^{13}C_{org}$ values of the alginite-rich layers (from −31.1 to −31.9‰) are similar to host rocks (from −30.5 to −31.9‰). The resulting values are typical for marine OM of the Bazhenov Formation that have been previously reported [4,62].

According to mineralogical studies, the main chemical form of sulfur in investigated samples is the sulfide sulfur fixed as pyrite. Organic sulfur presents in a very subordinate amount, the sulfur content in organic matter in sedimentary rocks of West Siberia is generally low and does not exceed 5 wt.% [62]. The sulfur isotope composition of sulfide sulfur indicates the changes in redox conditions during sedimentation and the sulfate-reducing bacteria activity. Intensive sulfate reduction in euxinic conditions results in higher sulfur content and its "lighter" isotope composition [63], which is observed in the host rocks with $\delta^{34}S_{tot}$ from −17.3 to −26.0‰. The alginite-rich layers are characterized by intermediate values of sulfur content and $\delta^{34}S_{tot}$ values from −6.0 to −18.5‰, which reflects suboxic conditions of sedimentation as compared with the host rocks. The tuff samples are quite different with low sulfur content and relatively high $\delta^{34}S_{tot}$ values from 2.4 to 4.5‰. The sulfur isotope composition of tuffs is possibly affected by the admixture of volcanic sulfur from pyroclastic material [64,65].

The low nitrogen concentration (0.07–0.21 wt.%), high $C_{org}/N_{tot}$ ratios (70–155 and above), and $\delta^{15}N_{tot}$ values (up to 25.7‰) are characteristic features of the alginite-rich layers. Host rocks have $C_{org}/N_{tot}$ ratios from 3 to 50 and $\delta^{15}N_{tot}$ values from 1 to 5.4‰ that show the same values as the black shales of various ages [66–68]. The low nitrogen content and heavy isotope composition correspond to an initial predominance of lipid components and/or significant microbial alteration of OM at the diagenesis stage. The process of bacterial transformation is known to be common for the sapropelic OM formation, and it is usually accompanied by destruction of proteins and carbohydrates [69]. The nitrogen isotope fractionation occurs due to amino acids destruction resulting in the loss of amino groups, and the kinetic isotope effect is directed towards the accumulation of lighter $^{14}N$ in the products of biochemical reactions and enrichment of the residual substrate by $^{15}N$ [70].

For comparison, luminescent tuffs exhibit relatively high nitrogen and low organic carbon concentrations, indicating the presence of inorganic nitrogen. In a detailed min-

eralogical study of tuff layers of the Bazhenov Formation, Shaldybin et al. found the illite/tobelite/smectite mixed-layered mineral compounds, where ammonium partially replaces potassium in the crystal structure of tobelite [10,71]. We attribute formation of these minerals to transformation of pyroclastic material at the diagenesis stage in the presence of ammonium, formed during the OM decomposition in the pore waters. High nitrogen content and low $\delta^{15}N_{tot}$ values, obtained for tuff samples in this study, are consistent with this concept.

### 5.3. Origin of the Alginite-Rich Layers

The obtained results show that extensive environmental changes have occurred, during the short term, in the geological timescale period of sedimentation of the paleobasin in the territory of modern Western Siberia. Due to these changes, a considerable amount of organic-matter-rich algal sapropel has accumulated in most parts of the paleosea. The observed thickness of lithified material corresponds to the sedimentation of 5–10 m of mud enriched with algae, which most probably was formed as a result of algal blooms.

Algae are known to occur in a variety of depositional environments and have been described from ancient deposits to recent times. The environment must have had a regular and ample nutrient supply, enough light intensity, and optimal water temperature. Algal blooms happen regularly in the Black and the Mediterranean Seas [72,73]. The source of nutrients in the middle of the paleosea potentially stimulating algal blooms could be volcanic ash [74–76], eolian processes [77–79], upwelling [80], or run-off from the eroding land areas, and currents, concentrating stock of living plankton. The occurrence of a large amount of quartz in the alginite-rich layers is associated with the release of silica or input from an external source. The source of silica most commonly includes biogenic debris (radiolarians, diatoms, sponge spicule, and others), the transformation of clay minerals, and the vitrification of volcanic ash. The aeolian processes can produce a large amount of quartz-rich dust from coastal deserts [81,82].

### 5.4. Stratigraphy and Hydrocarbon Potential

In the Bazhenov Formation, the luminescent alginite-rich layers occur in addition to luminescent tuffs in Unit 4 (Figure 11). They have different types of luminescence and can be distinguished by the criteria listed above. The widespread distribution over the entire area of the central part of Western Siberia makes both the alginite-rich layers and tuffs useful for correlating the Bazhenov sequence.

Luminescence of the alginite-rich layers is associated with a significant amount of luminescent alginite in their composition. During the study, we did not found any luminescent alginite-rich layers in wells with the degree of OM maturity after the middle of the oil window. This observation allows us to suggest that at a certain thermal maturity degree during realization of the generation potential of OM, the luminescent alginite-rich layers lose their luminescence. Speight and Huc described samples of type I kerogen that contained algobacterial material that luminescent under UV light at low thermal maturity stages, and the intensity of luminescence gradually decreased and disappeared with maturation [83,84]. Therefore, since OM reached the middle of the oil window, the luminescence of the alginite-rich layers was extinguished and the layers could be distinguished by geochemical data.

Luminescence of the alginite-rich layers can be used to confirm the thermal maturity parameters of Rock-Eval pyrolysis or biomarker analysis of Bazhenov deposits from immature conditions (PC3) to the early to middle oil window (MC1-2). Generally, the luminescent colors of the alginite-rich layers change from a greenish-yellow color to a bright yellow and orange under UV light before luminescence is mostly extinguished at the middle oil window (Figure 12). The decreasing alginite luminescence of high maturity level is due to a decrease in the ratios of both hydrogen and oxygen to carbon and increased molecular aromatization [85–88].

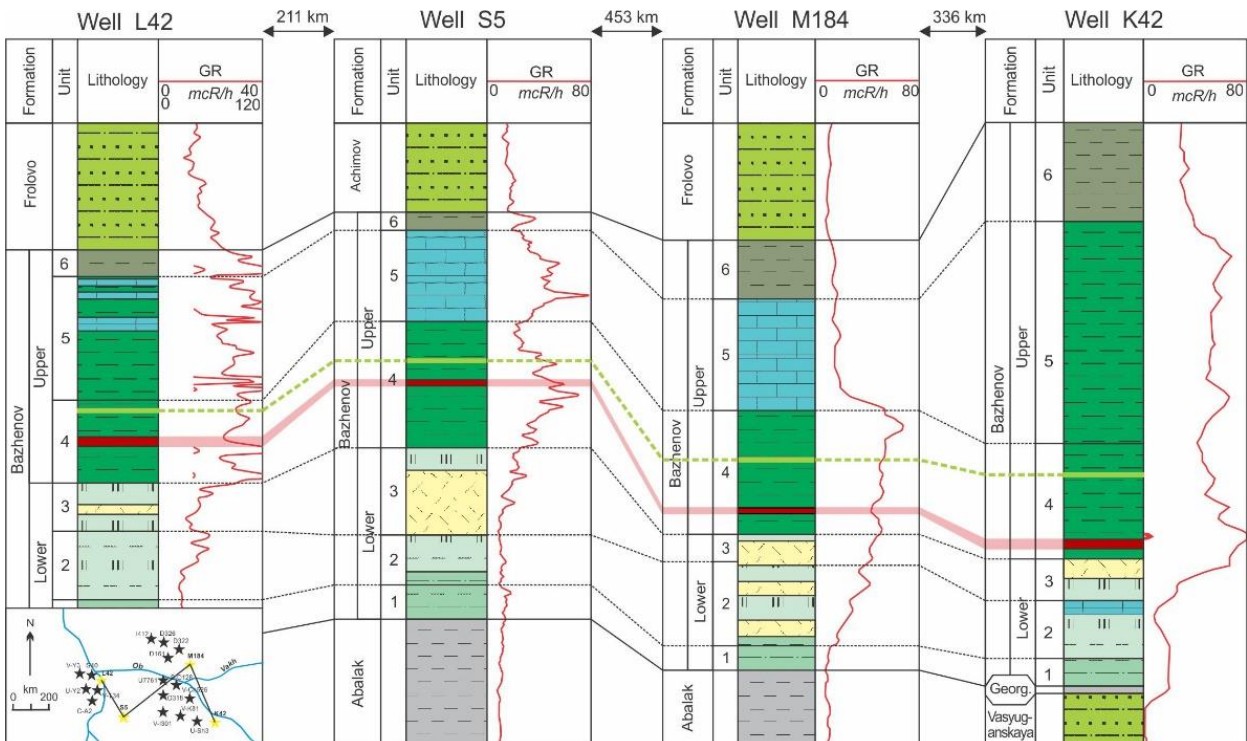

**Figure 11.** Stratigraphic cross-section showing the position of the alginite-rich layers (marked in red) and tuffs (marked in light green) throughout the Bazhenov sequence. Location of the well and correlation line are in the map. The legends of lithology are the same as those in Figure 2. GR, gamma-ray log.

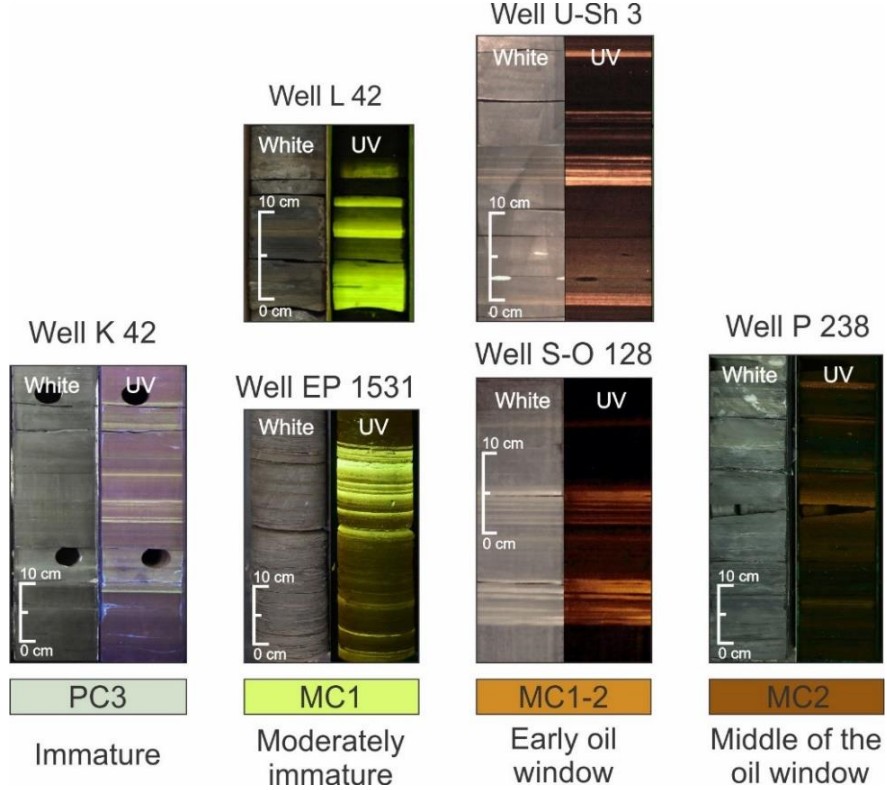

**Figure 12.** Color of the luminescent alginite-rich layers in the Bazhenov deposits of different thermal maturity.

The hydrocarbon generation potential of the alginite-rich layers is significantly higher than that in the host rocks enriched in type II kerogen. The value of S2 for the alginite-rich layers is 2–2.5 times higher than that in the host rocks and reaches 250 mg HC/g rocks. The prevalence of aliphatic compounds in generated hydrocarbons by type I kerogen contributes to the high quality of the produced crude oils. Therefore, the alginite-rich layers improve the petroleum generation potential of the Bazhenov Formation and are suggested to be significant contributors to hydrocarbon formation. Since type II kerogen exhausted its petroleum generation potential at a high maturity level of the Bazhenov Formation, the type I kerogen of the alginite-rich layers still holds its high hydrocarbon potential and is capable of producing light oils. In the oil fields within the Western Siberian basin with a high OM thermal maturity (Verkhnesalymskoye, Krasnoleninskoye, and Sredne-Nazymskoye) type I kerogen is possibly the major source of petroleum derived from the Bazhenov strata.

A high content of type I kerogen is concentrated in luminescent Stellarites of Late Carboniferous within the Stellarton basin in Canada [89,90]. Algal-rich sediments of the Late Jurassic–Early Cretaceous age have been found in the southern part of the Barents Sea [91,92]. The Blackstone layer of the Late Jurassic Kimmeridge Formation consists primarily of algal origin OM [93,94]. Other records of abundant fossil algae confirm that they can be important contributors of OM to source rock deposits.

## 6. Conclusions

The luminescent alginite-rich layers were discovered in many wells of the Western Siberia petroleum basin, distributed over an area of 0.5 million km$^2$. They were situated stratigraphically in the Upper part of the Bazhenov Formation. Individual laminas have thicknesses from 1 to 50 mm, and the total thickness of the layers in one cross-section reaches 1 m.

We determined the key characteristics during the comparative analysis of the alginite-rich layers with host rocks and luminescent tuffs. For the alginite-rich layers, the luminescence under UV light was associated with algal OM. Abnormally high HI values, high atomic H/C, and the single activation energy of thermal decomposition, which are typical for type I kerogen, were obtained for the alginite-rich layers. The stable isotope composition of carbon has similar values for the alginite-rich layers and host rocks, whereas tuffs have higher and more positive values. More significant differences were found in the elemental and isotope composition of nitrogen and sulfur as compared with the host rocks and tuffs.

The obtained results show that extensive environmental changes have occurred, during the short term, in the geological timescale period of sedimentation of the paleobasin in the territory of modern Western Siberia. Due to these changes, a considerable amount of algal-rich OM has accumulated in most parts of the paleosea, most probably as a result of algal blooms.

For practical purposes, the described alginite-rich layers can become a tool for correlation of the Bazhenov sequence. The exceptionally high generation potential of the alginite-rich layers organic matter and the composition of thermal destruction products should be taken into consideration during basin modeling of the Western Siberian petroleum basin.

**Author Contributions:** Conceptualization, methodology, and manuscript preparation, T.B., E.K. and E.L.; supervision, E.K., M.S. and N.M.; petrographic, XRD, and XRF analysis, T.B.; Rock-Eval pyrolysis, bulk kinetics, and elemental analysis, E.K. and T.B.; maceral analysis, N.P.; pyro-GC×GC-TOF MS/FID, E.L.; isotope analysis, A.V.; writing—original draft preparation, T.B., E.K., E.L. and M.S.; writing—review and editing, T.B., E.K., M.S., E.L., A.V., I.P., N.P. and N.M. All authors have read and agreed to the published version of the manuscript.

**Funding:** This work was supported by the Ministry of Science and Higher Education of the Russian Federation under agreement No. 075-10-2020-119 within the framework of the development program for a world-class research center.

**Institutional Review Board Statement:** Not applicable.

**Informed Consent Statement:** Not applicable.

**Acknowledgments:** The authors would like to thank the Center for Hydrocarbon Recovery Laboratory members for technical support. The authors thank anonymous reviewers, whose constructive comments and suggestions greatly improved this manuscript.

**Conflicts of Interest:** The authors declare no conflict of interest.

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
