# Peer review of "Alginite-Rich Layers in the Bazhenov Deposits of Western Siberia"

_geosciences, doi:10.3390/geosciences11060252_

Round 1
Reviewer 1 Report
In my opinion, the manuscript is very interesting and can be recommended for publication. The paper presents the results of the study of the important object for oil and gas geology - the Bazhenov formation. The presence of alginite interlayers related to type 1 kerogen has been proven (earlier the kerogen of the Bazhenoskaya suite was referred to type 2). A positive fact is a huge number of reference to Russian-language sources that are inaccessible to non-Russian-speaking scientists.
The studies described in the work are complex: a significant amount of core material sampled from various areas of the Bazhenov formation has been studied by various methods.
In my opinion, the manuscript contains some inaccuracies that should be corrected before publication.
Row 72: “CHNS analysis” : This analysis is not described in the Materials and Methods section.
Row 84: “The sedimentation period covers the Tithonian-early Valanginian time” - It is more correct to write “late Tithonian – early Valanginian time”.
Figure 2. It is necessary to clarify the following terms in the legend: BK, GR, IK, NKTb. For core photographs, in the legend it is necessary to indicate which photograph is without ultraviolet light (left) and which one is with ultraviolet light (right).
Rows 109-110: “There are minor quantities of siliceous rocks composed of radiolarites and carbonate rocks (limestones and dolomites).” - However, judging by Figure 2, the amount of carbonates is quite large - see unit 5 (number 3 in legend).
Rows 124-125: You must specify that MS - means mesocatagenesis and PC – protocatagenesis
Row 134: “The are other….” - Is there a typo in this sentence?
Row 166: You are listing the pyrolysis parameters of S0 - S5, however, further in the text you do not discuss S3-5 in any way. Explain the meaning of S4 and S5.
Rows 195-195: “The individual alginate-rich layer thickness varies….” - Indicate the thickness of the entire studied cross-section.
Figure 4: Perhaps, the main elements (organic matter, quartz, volcanic ash, pyrite, radiolaria, etc.) should be indicated directly on the micrographs with some symbols?
Rows 331-332: “Results of pyrolysis-gas chromatography/mass spectrometry revealed data of the kerogen types presented in the alginate-rich layers and host rocks (table 2).” - Explain in more details the meaning of the values of the parameters indicated in Table 2. What do these values indicate? Why are you showing them in the manuscript? Explain these results in the text.
Row 410: Insert a “minus” in front of isotopic value 17.3.
Rows 428-431: “In detailed mineralogical study of tuff layers of the Bazhenov Formation, Shaldybin and co-authors have found the illite/tobelite/smectite mixed-layered mineral compounds……” Based on this sentence, we can conclude that smectite is an important component composing the tuff layers. Personally, I fully understand this process and agree with it. However, above, in lines 244-245, you indicated that the tuff interlayers are composed of mixed-layered minerals and kaolinite. I don't really understand how kaolinite could be obtained from these tuffs? Are these tuffs acidic? But Shaldybin points to the presence of smectites and does not write anything about kaolinite. It seems to me that there is a contradiction here, please, explain it. At the end of the sentence, you mistakenly put a reference to work 67, authored by Galimov. The correct reference is 69.
Row 441: “Observed thickness of lithified material corresponds to the sedimentation of 5-10 m of silt….” Why Silt? Have you done grain-size analysis? If not, add a reference.
Figure 12. In the figure caption, explain the difference between the left and right cores.
Rows 512-514: “Significant differences were found in elemental and isotopic composition of carbon, nitrogen and sulfur compared to host rocks and luminescent tuffs.” - However, there is no difference in the isotopic composition of carbon, which you rightly wrote above in the corresponding section of the manuscript.
Author Response
Dear Reviewer, thank you for your insightful comments!
Please see the attachment.
Best regards,
Timur Bulatov

Reviewer 2 Report
Please, see my comments and suggestions in the attached file.

Author Response
Dear Reviewer,
Thank you for your consideration! We appreciate your insightful comments!
We accepted all your suggestions that greatly improved this manuscript.
I am really sorry for confusing you with the Russian abbreviation of well log data. In figure 2 and in the caption we corrected it. Legend for the lithological column we also put in figure 2 caption.
For thermal maturity nomenclature in this publication, we use international terms (oil window) and widely accepted in Russia abbreviations (PC, MC) as well.
PC (protocatagenesis) is the substage of catagenesis that corresponds to the immature organic matter. Protocatagenesis is also subdivided into PC1, PC2, and PC3.
MC (mesocatagenesis) is also the substage of catagenesis that corresponds to more mature organic matter. Mesocatagenesis subdivided into MC1 (moderately immature OM), MC1-2 (early oil window), MC2 (middle of the oil window), MC3(end of the oil window), MC4, and MC5 (gas window).
We described in more detail the thin section petrography and organic petrography that we used in our study. Rock-Eval parameters we described more completely as well in the Materials and Methods section.
The source of luminescence in the tuffs is still unclear. In previous publications, authors associate luminescence with clay mineralogy (Shaldybin et al., 2019 [10 in References]) or concentration of barite (Panchenko et al., 2021 [11 in References]). We mentioned it in our manuscript in the Geological Setting section.
Best regards,
Timur Bulatov

Reviewer 3 Report
Very great job, everything looks perfect. Thanks for your quality job.
Author Response
Dear Reviewer,
Thank you for your consideration!
Best regards,
Timur Bulatov